# Cigarette smoking and associated factors among men in five South Asian countries: A pooled analysis of nationally representative surveys

Md Shariful Islam[1]*, Mamunur Rashid[2], Monaemul Islam Sizear[3,4], Raafat Hassan[5], Mahbubur Rahman[1], Sarker Masud Parvez[1,6], Shuvon Chandra Hore[7], Rehnuma Haque[1], Farjana Jahan[1], Supta Chowdhury[8], Tarique Mohammad Nurul Huda[1], K. M. Saif-Ur-Rahman[9], Arifuzzaman Khan[6,10]

1 Infectious Diseases Division, icddr,b, Dhaka, Bangladesh, 2 Department of Public Health and Sports Science, Faculty of Health and Occupational Studies, University of Gävle, Gävle, Sweden, 3 Public Health Foundation, Bangladesh, Dhaka, Bangladesh, 4 ThinkWell, Dhaka, Bangladesh, 5 BRAC James P Grant School of Public Health, BRAC University, Dhaka, Bangladesh, 6 Central Queensland Public Health Unit, Central Queensland Hospital and Health Service, Rockhampton, Queensland, Australia, 7 Dhaka Metropolitan Police, Dhaka, Bangladesh, 8 National Nutrition Services, Ministry of Health and Family Welfare, Dhaka, Bangladesh, 9 Health Systems and Population Studies Division, icddr'b, Dhaka, Bangladesh, 10 Faculty of Medicine, The University of Queensland, Brisbane, Queensland, Australia

* shariful.islam4@icddrb.org

**Data Availability Statement:** All relevant data are within the manuscript and its Supporting Information files.

## Abstract

Smoking is one of the leading causes of premature deaths worldwide. The cigarette is the commonest form of tobacco smoking. This study investigated the factors associated with cigarette smoking among men in five South Asian countries. We analyzed nationally representative cross-sectional study (Demographic and Health Survey) data conducted in Afghanistan, India, Maldives, Nepal, and Pakistan from 2015–2018. Our study population was men aged between 15 and 49 years. The outcome variable was the prevalence of cigarette smoking. We performed both pooled and country-specific analyses using multivariable logistic regression. The prevalence of cigarette smoking among men is the highest (41.2%) in the Maldives and the lowest (20.1%) in Pakistan. Our pooled analysis found that higher age, lower education, lower wealth status, and involvement in any occupations were strongly associated with cigarette smoking (*p*-value <0.001). However, we did not find a significant association between age and wealth status in Afghanistan, occupations in Nepal and Pakistan, and education in Pakistan with cigarette smoking when country-specific analyses were performed. In this study, socioeconomic position, age, and urban area are strongly associated with cigarette smoking in South Asian countries. The country-specific circumstances should be considered in planning and designing national smoking control strategies and interventions. However, improving access to smoking cessation services could be an effective intervention for all studied countries, Afghanistan, India, Maldives, Nepal, and Pakistan.

**Funding:** The author(s) received no specific funding for this work.

**Competing interests:** The authors have declared that no competing interests exist.

**Abbreviations:** AOR, Adjusted Odds Ratio; CI, Confidence Interval; DHS, Demographic and Health Survey; UOR, Unadjusted Odds Ratio.

## Introduction

Tobacco use is a major preventable cause of morbidity, mortality, and impoverishment in the world [1]. In the twentieth century, tobacco use caused around 100 million deaths worldwide, most of which occurred in developed countries [2, 3]. If existing smoking habits continue, tobacco will kill about one billion people this century, the majority of whom will be in the low- and middle-income countries [2–5]. The prevalence of tobacco smoking is growing, especially in lower-income countries [6]. A total of 80% of global tobacco smokers live in low and middle-income countries [1]. In South Asia, smoking prevalence is estimated as 25·2% among men and 3·26% among women [7]. This generates a severe public health concern and a crucial modifiable risk factor for leading non-communicable diseases in this region [8]. More than one million people die every year in the South and Southeast Asian region due to tobacco smoking, which is significantly higher than in any other region [9]. The economic cost of smoking is even higher in this region, estimated at 319 billion PPP dollars annually [10]. The World Health Organization (WHO) introduced a target to reduce 25% of death from cardiovascular diseases, diabetes, cancer, and chronic respiratory diseases among individuals aged 30–70 years between 2010 and 2025 [11]. To achieve the target of non-communicable diseases in South Asian countries, reducing cigarette use can be the best single prevention and cost-effective approach.

In South Asia, both tobacco smoking (25.2%) [7] and smokeless tobacco (24.7%) [12] are popular among men. However, studies on factors associated with cigarette smoking among men and based on nationally representative and multi-country data are sparse in South Asia. Some single-country studies have identified the factors associated with all tobacco use in Nepal [13] and India [14]. Those studies reported that education status, married men, lower wealth status, and poor access to media were associated with smoking. Men who lived in urban areas, from lower wealth status and manual workers, consumed smoked tobacco more than smokeless tobacco in India [14]. The prevalence of cigarette smoking was higher among men from the richest household, whereas smokeless tobacco use was more common among the poorest households in Afghanistan [15]. In South Asian countries, various forms of tobacco smoking products are available such as manufactured cigarettes, bidis (hand-roll cigarette), hookah, cigars, pipes, kreteks, sheesha, sulpha, chilam, and so on. Cigarette smoking is the most popular form of smoking in this region [16–18]. However, no study examined the factors associated with cigarette smoking in men in South Asian countries, while more than one-fifth of the total global smokers (15 years and above) are from this region [19]. Due to the higher burden of cigarette smoking than smokeless tobacco among men, it is essential to know the prevalence and associated factors in South Asian countries. Alternatively, primary prevention initiatives in smoking control in South Asia are limited by the government and other private organizations [20]. We have selected five South Asian countries to identify associated factors of cigarette smoking among men (15–49 years). This study can generate evidence that would help to develop evidence-based policy.

## Methods

### Data source and study design

We conducted secondary data analysis and drew the data from the latest Demographic and Health Survey (DHS) (It is known as National Health and Family Survey (NHFS) in India) from the five South Asian countries carried out in Afghanistan (2015), India (2016), Maldives (2017), Nepal (2016), and Pakistan (2018). These are nationally representative surveys, and the main objective was to collect updated information following the country's required health and demographic indicators. A stratified two-stage sample design was used in these cross-sectional

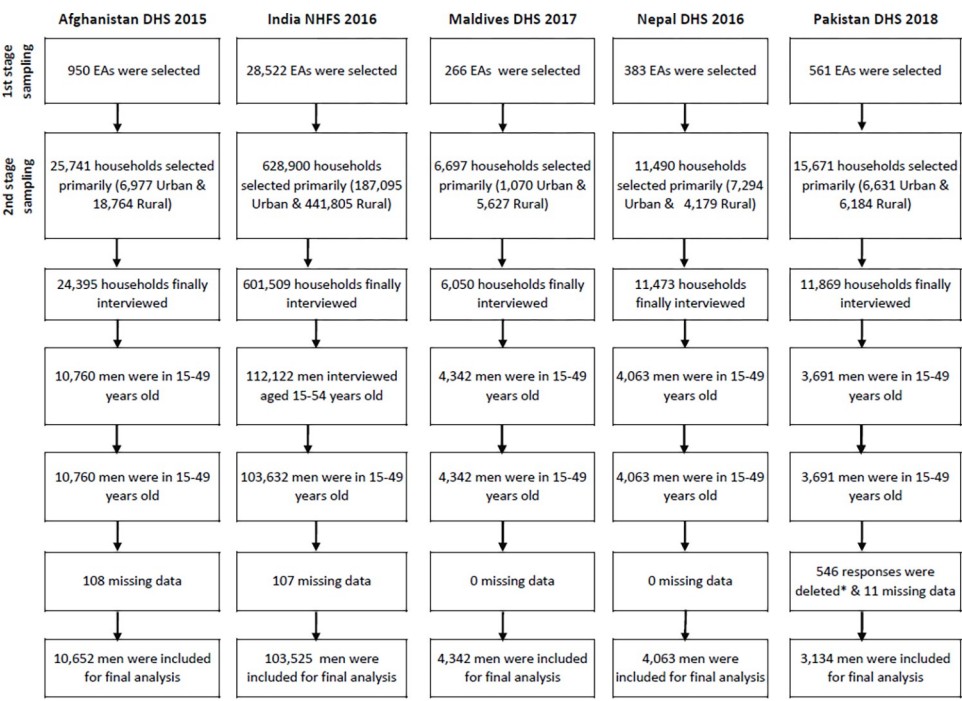

**Fig 1. Flowchart of sample selection.**

studies. Each province/states is divided into urban and rural areas to achieve stratification. In the first stage of sampling, enumeration areas/blocks were selected from the urban and rural areas of each province. The number of enumeration areas selected depends on the population size of the province. DHS used enumeration areas of the last census of respective countries (see Fig 1). A fixed number of households were identified from each enumeration area in the second stage. The DHS research team calculated the sample size. Sample sizes for these DHS surveys are based on the number of survey domains (usually subnational units such as regions). All survey sampling strategies are subject to sampling error. The DHS Program designs samples to provide national and subnational estimates with a reasonable relative standard error. During collecting data for the DHS survey, trained data collectors performed face-to-face interviews using a standardized questionnaire incorporating reliable and valid instruments. The detailed sampling strategy and data collection are given in the DHS reports [15–18, 21]. The Strengthening the Reporting of Observational Studies in Epidemiology (STROBE) statement was followed in this research [22].

## Inclusion criteria

Study participants were men who live in Afghanistan, India, Maldives, Nepal and Pakistan. The age of participants was between 15 and 49 years old. We excluded women because the prevalence of cigarette smoking among women in South Asian countries is very low.

## Study setting and population

The study participants were men aged between 15 and 49 from the five South Asian countries (Afghanistan, India, Maldives, Nepal, and Pakistan). South Asia includes the countries: Afghanistan, Bangladesh, Bhutan, India, Maldives, Nepal, Pakistan, and Sri Lanka. DHS data

on cigarette smoking was unavailable in Bangladesh, Bhutan, and Srilanka. Data from five countries have been analyzed in this study that is representative of 89.5% of the total population aged 15–49 years in South Asia [23]. The overall response rate was approximately 92%.

## Outcome

The outcome of the study was the prevalence of cigarette smoking. Respondents were asked, "Does he currently smoke cigarettes" and the response was taken as "Yes" or "No". Consuming any cigarettes within the last seven days is counted as currently smoking cigarettes. This included both manufactured and hand-rolled cigarettes in all countries.

## Independent variables

We selected factors of cigarette smoking based on previous research conducted on tobacco use in South Asia [13, 14, 24]. The factors included in this study were age, marital status, residence, region, education, occupation, wealth, watching television, listening to the radio, and reading newspapers. Education level was categorized into four groups, no education (0 schooling years), primary (1–5 schooling years), secondary (6–12 schooling years), and higher (12 + schooling years). Similarly, occupation status was grouped into four groups (not working, agriculture, skilled/unskilled manual, and professional/technical/managerial/services). We grouped household wealth status into five indexes (poorest, poorer, middle, richer, and richest). Household wealth status was classified by the DHS program, which was specific for each country and based on household assets and dwelling characteristics [25]. Access to the information, newspaper/magazine, television, and radio all was categorized into three groups (not used at all, less than once a week, and at least once a week). Age was categorized into four groups (15–19, 20–29, 30–39, and 40–49). Marital or union status was grouped into currently married or union, never married or not union, and formally married or union. The place of residence was divided into rural and urban areas. Regions were named as the state names in all countries except in Nepal, where it is known as a province. S1 Table summarizes the independent variables and the methods used to measure them.

## Statistical analysis

We conducted descriptive statistics and reported them as frequencies and proportions. The prevalence of cigarette smoking among different selected variables was calculated separately in each country. We carried out the chi-square test to determine the association of factors with cigarette smoking. We performed multivariable logistic regression for each country separately. We also calculated multivariate logistic regression in a pooled dataset of five countries using the country of domicile as dummy-variable controls. We checked the multicollinearity of independent variables using variation inflation factors (VIF). The association of factors with cigarette smoking was expressed as an adjusted odds ratio (AOR) with 95% confidence intervals (95% CIs), with a p-value of less than 0.05 considered significant. All analyses were two-tailed. The complex sampling design and sampling weight were considered in the prevalence and multivariable regression model. According to the DHS's recommendation, sampling weight was re-normalized during pooled analysis [26]. The final analysis excluded missing values. The statistical software R 4.0 was used to analyze the data.

## Ethical consideration

This secondary data analysis did not receive any additional ethical clearance. The Demographic Health Survey program was approved by ethical review boards of the respective

authorities before conducting the primary study. All DHS surveys have been ethically approved by ICF International as well as a national institutional review Board from each country (Afghanistan, Nepal, India, Pakistan, and the Maldives) should be in agreement with the U. S. Department of Health and Human Services regulations for the protection of human subjects (https://www.dhsprogram.com).

## Results

### Socio-demographic characteristics of participants

The highest number of participants (103,525) in this study was from India (Table 1). The percentage of responders aged 20–29 years old was the highest in India (31.7%), Nepal (28.4%), Maldives (31.0%), whereas a larger proportion of the participants in Pakistan (40.8%) and Afghanistan (36.0%) were 30–39 years olds. The majority of respondents were currently married or in union. The larger part of the participants lived in the rural area in India (68.3%), Maldives (85.5%), and Afghanistan (78.3%); however, a higher percentage in Pakistan (52.2%) and Nepal (65.6%) were from urban area. In Afghanistan, 51.4% of participants had no education, while in the other four countries, a higher proportion of respondents had secondary education levels. A total of 40.3% of respondents in Pakistan, 45.0% in the Maldives, 33.0% in Afghanistan, and 31.2% in Nepal were involved in professional/technical/managerial/services, while 28.4% of men in India were engaged in agriculture. The number of participants was mainly distributed similarly in the five wealth index in all countries, except in Afghanistan and Maldives, where 14.0% and 7.3% of respondents were in the richest index (Table 1).

Cigarette smoking among men varied remarkably among the five countries, the highest in the Maldives (41.2%, 95% CI 39.2–43.2) and the lowest in Pakistan (20.1%). In Nepal, around one in three men smoked cigarettes, while the prevalence in Afghanistan and India was 22.0% and 23.0%, respectively (Fig 2) (S2 Table). Prevalence variation of cigarettes across all states of five countries was high, 72.1% in Mizoram of India to 2.5% in Nimroz of Afghanistan. The highest proportion of cigarette smoking in Afghanistan was in Jawjan state (49.5%). Province 7 (39.5%) in Nepal, Islamabad (26.6%) in Pakistan, Central (45.8%) region of Maldives had the highest proportion of cigarette smoking among men (Fig 3) (S3 Table).

The prevalence of cigarette smoking increased with the increased age of men in South Asian countries, except in the Maldives. In the Maldives, the highest proportion of cigarette smoking was seen among men aged 20–29 years old (52.7%; 95% CI 49.6–55.9). In all South Asian nations except Pakistan, men who were formally married or in union smoked more cigarettes than men who were currently married or had never been married. The proportion of cigarette smoking decreased among men who have schooling more than the primary level in South Asia, except for Afghanistan (Table 2). Cigarette smoking prevalence was the lowest among men who were not working in all five South Asian countries. Men who worked in agriculture had the highest prevalence of cigarette smoking in Pakistan (30.4%, 95% CI 25.5–35.5) and Maldives (51.0%, 95% CI 47.0–55.0), and among men who worked in skilled/unskilled manual sectors in Nepal (35.0%, 95% CI 30.9–39.2), in India (29.6, 95% CI 28.6–30.6), and Afghanistan (23.6%, 95% CI 20.8–26.6). Men who were from the poorest wealth quintile among the five wealth status in Nepal (34.0%), Pakistan (25.8%), India (29.1%), and Maldives (46.0%) had the highest prevalence of cigarette smoking. The prevalence of cigarette smoking decreased with an increase in information access in India and the Maldives.

### Associated factors with cigarette smoking

**Pooled analyses.** The magnitude of association of cigarette smoking increased with age in South Asian countries. The adjusted odds ratio among men aged 40–49 years was 4.61

**Table 1. Characteristics of the study population[a].**

| Factors | Afghanistan | India | Maldives | Nepal | Pakistan | Overall[b] |
|---|---|---|---|---|---|---|
| | N(%)[c] | N(%)[c] | N(%)[c] | N(%)[c] | N(%)[c] | N(%)[c] |
| | 10652 (100) | 103525 (100) | 4342 (100) | 4063 (100) | 3134 (100) | 125716 (100) |
| **Age** | | | | | | |
| 15–19 | 158 (1.5%) | 19082 (18.4) | 950 (21.9) | 964 (23.7) | 48 (1.5) | 21202 (16.9) |
| 20–29 | 3625 (34.0) | 32781 (31.7) | 1347 (31.0) | 1155 (28.4) | 848 (27.1) | 39756 (31.6) |
| 30–39 | 3831 (36.0) | 28537 (27.6) | 1162 (26.8) | 1048 (25.8) | 1278 (40.8) | 35856 (28.5) |
| 40–49 | 3038 (28.5) | 23125 (22.3) | 883 (20.3) | 896 (22.1) | 960 (30.6) | 28902 (23.0) |
| **Marital or Union status** | | | | | | |
| Never Married or Union | NA | 40136 (38.8) | 1750 (40.3) | 1341 (33.0) | NA | 43227 (34.4) |
| Currently Married or Union | 10581 (99.3) | 62091 (60.0%) | 2418 (55.7) | 2691 (66.2) | 3080 (98.3) | 80861 (64.3) |
| Formally Married or Union | 71 (0.7) | 1298 (1.3) | 174 (4.0) | 31 (0.8) | 54 (1.7) | 1628 (1.3) |
| **Place of Residence** | | | | | | |
| Urban | 2311 (21.7) | 32771 (31.7) | 628 (14.5) | 2667 (65.6) | 1636 (52.2) | 40013 (31.8) |
| Rural | 8341 (78.3) | 70754 (68.3) | 3714 (85.5) | 1396 (34.4) | 1498 (47.8) | 85703 (68.2) |
| **Education** | | | | | | |
| Higher | 774 (7.3) | 16542 (16.0) | 622 (14.3) | 838 (20.6) | 721 (23.0) | 19497 (15.5) |
| Secondary | 2681 (25.2) | 61706 (59.6) | 2570 (59.2) | 2034 (50.1) | 1072 (34.2) | 70063 (55.7) |
| Primary | 1724 (16.2) | 12684 (12.3) | 1010 (23.3) | 790 (19.4) | 543 (17.3) | 16751 (13.3) |
| No Education | 5473 (51.4) | 12593 (12.2) | 140 (3.2) | 401 (9.9) | 798 (25.5) | 19405 (15.4) |
| **Wealth** | | | | | | |
| Poorest | 1944 (18.3) | 17035 (16.5) | 1127 (26.0) | 778 (19.1) | 578 (18.4) | 21462 (17.1) |
| Poorer | 2451 (23.0) | 21584 (20.8) | 1141 (26.3) | 789 (19.4) | 643 (20.5) | 26608 (21.2) |
| Middle | 2404 (22.6) | 22604 (21.8) | 1217 (28.0) | 797 (19.6) | 569 (18.2) | 27591 (21.9) |
| Richer | 2363 (22.2) | 21516 (20.8) | 542 (12.5) | 896 (22.1) | 654 (20.9) | 25971 (20.7) |
| Richest | 1490 (14.0) | 20786 (20.1) | 315 (7.3) | 803 (19.8) | 690 (22.0) | 24084 (19.2) |
| **Occupation** | | | | | | |
| Not working | 346 (3.2) | 24170 (23.3) | 723 (16.7) | 587 (14.4) | 97 (3.1) | 25923 (20.6) |
| Professional/technical/managerial/services | 3516 (33.0) | 23809 (23.0) | 1952 (45.0) | 1268 (31.2) | 1263 (40.3) | 31808 (25.3) |
| Agriculture | 3357 (31.5) | 29439 (28.4) | 755 (17.4) | 1262 (31.1) | 548 (17.5) | 35361 (28.1) |
| Skilled/unskilled manual | 3433 (32.2) | 26107 (25.2) | 912 (21.0) | 946 (23.3) | 1226 (39.1) | 32624 (26.0) |
| **Access to the information** | | | | | | |
| **Frequency of watching television** | | | | | | |
| Not at all | 5131 (48.2) | 15150 (14.6) | 327 (7.5) | 892 (22.0) | 895 (28.6) | 22395 (17.8) |
| Less than once a week | 1124 (10.6) | 10858 (10.5) | 677 (15.6) | 1231 (30.3) | 535 (17.1) | 14425 (11.5) |
| At least once a week | 4397 (41.3) | 77517 (74.9) | 3338 (76.9) | 1940 (47.7) | 1704 (54.4) | 88896 (70.7) |
| **Reading newspaper or magazine** | | | | | | |
| Not at all | 8274 (77.7) | 34548 (33.4) | 1273 (29.3) | 1911 (47.0) | 1695 (54.1) | 47701 (37.9) |
| Less than once a week | 1032 (9.7) | 16659 (16.1) | 735 (16.9) | 1428 (35.1) | 564 (18.0) | 20418 (16.2) |
| At least once a week | 1346 (12.6) | 52318 (50.5) | 2334 (53.8) | 724 (17.8) | 875 (27.9) | 57597 (45.8) |
| **Frequency of listening to the radio** | | | | | | |
| Not at all | 3625 (34.0) | 73009 (70.5) | 2455 (56.5) | 1081 (26.6) | 2414 (77.0) | 82584 (65.7) |
| Less than once a week | 2034 (19.1) | 9894 (9.6) | 865 (19.9) | 1484 (36.5) | 408 (13.0) | 14685 (11.7) |

(*Continued*)

**Table 1.** (Continued)

| Factors | Afghanistan | India | Maldives | Nepal | Pakistan | Overall[b] |
|---|---|---|---|---|---|---|
| At least once a week | 4993 (46.9) | 20622 (19.9) | 1022 (23.5) | 1498 (36.9) | 312 (10.0) | 28447 (22.6) |

[a] The data comes from Standard Demographic and Health Survey (DHS) carried out in Nepal in 2016, Pakistan in 2018, the Maldives in 2017, Afghanistan in 2015, and from National Family Health Survey conducted in India in 2016. Data are not weighted in this table.

[b] Pool of all five countries.

[c] Column percentages

N–number of respondents

NA.–Not Applicable

($p<0.001$) times higher compared to 15–19 years olds. Urban men were 12% more likely to be smokers than rural men. Higher education and household wealth acted as protective factors; however, involvement in any work increases the chance of being a cigarette smoker in South Asian countries. The odds ratio was 2.20 (AOR 95% CI 1.93–2.49) among men with no education and 0.61 (AOR 95% CI 0.54–0.69) among men from the richest households. Men who had manual work had a 55% higher chance of being cigarette smokers than those who did not involve in any work. Access to newspapers or magazines at least once a week acted as a

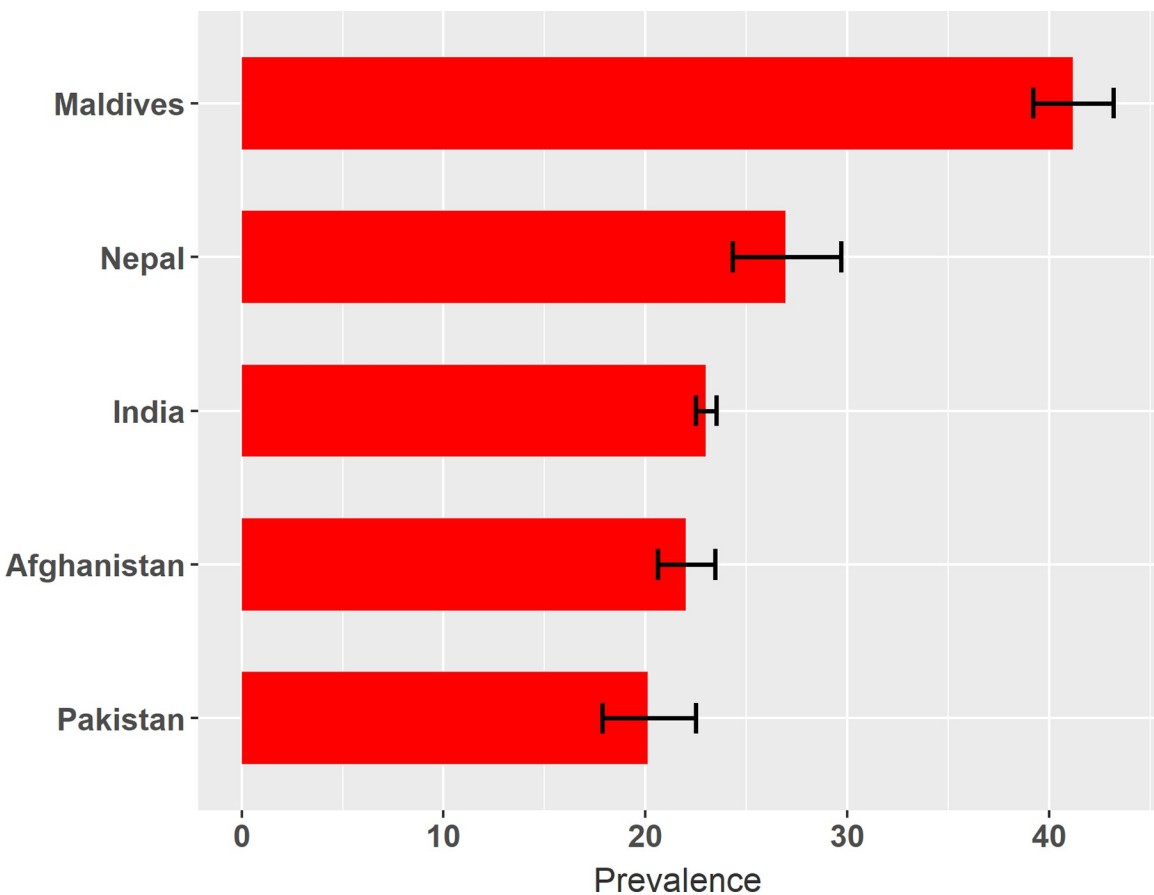

**Fig 2. National level prevalence of cigarettes among men in five South Asian countries (N = 125,716).** Prevalence is shown as a percentage with a 95% confidence interval (CI) value. The black-coloured error bar shows 95% CI. The countries were ranked based on the prevalence of cigarette smoking among men aged 15–49 years.

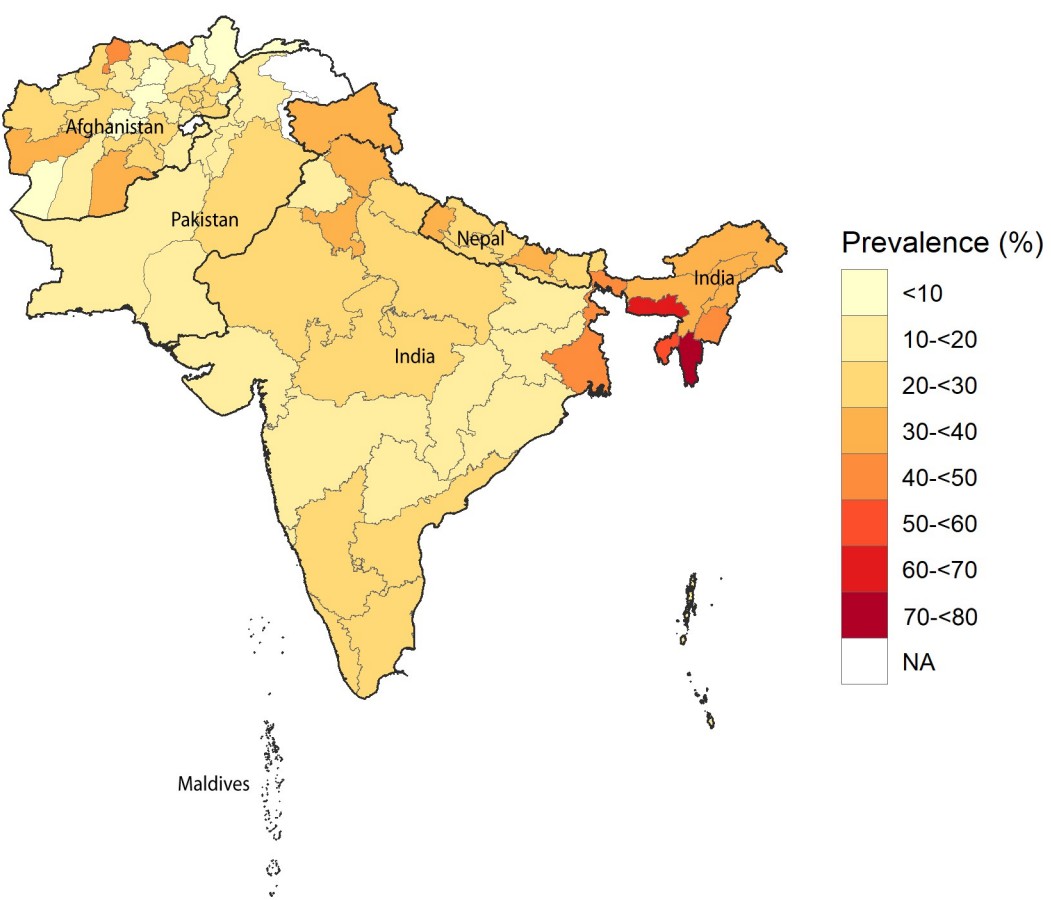

**Fig 3. State–level prevalence of cigarettes in the five South Asian countries.** Prevalence is shown in percentage where light colour showed a low prevalence and dark colour showed high prevalence. The map's thick black lines mark national borders, and thin grey lines mark the countries' first sub-national/ state borders.

protective factor against smoking cigarettes. In South Asian countries, however, the availability of other media (television or radio) enhanced the likelihood of being a cigarette smoker (Fig 4).

**Country–specific analyses.** *Afghanistan.* Men with education less than a higher degree had more chance of smoking cigarettes than those who were of higher education (AOR 2.37; 95% CI 1.35–4.14 in no education) in Afghanistan. There was a significant association between working status and cigarette smoking. Reading newspapers or magazines at least once a week acted as a protective factor among Afghan men, while watching television at least once a week increased the chance of being a smoker. The factors–listening to the radio, age, wealth status, marital status, and the residing area were not significantly associated with cigarette smoking among men in Afghanistan (Table 3).

*India.* The strongest association was found between cigarette smoking and age among the selected variables, and the odds of cigarette smoking increased with age in India. The AOR was 4.40 times among men aged 40–49 years old than the men aged 15–19 years. Higher wealth quintile and education had a protective effect on smokers among men in India. However, men who worked in any occupations were more likely to smoke cigarettes than men who did not work. Residing in urban areas, formally married men, access to television and radio increases the chance of being cigarette smokers, while reading newspapers or magazines played a protective role in being smokers in India (Table 3).

**Table 2.  Prevalence of cigarettes smoking among men in five South Asian countries[a].**

| Factors | Afghanistan | India | Maldives | Nepal | Pakistan |
|---|---|---|---|---|---|
| | P–value | P–value | P–value | P–value | P–value |
| | Prevalence (%) | Prevalence (%) | Prevalence (%) | Prevalence (%) | Prevalence (%) |
| | (95% CI)[b] | (95% CI)[b] | (95% CI)[b] | (95% CI)[b] | (95% CI)[b] |
| **Age** | 0.048 | <0.001 | <0.001 | <0.001 | <0.001 |
| 15–19 | 13.9 (7.7–22.2) | 7.8 (7.2–8.4) | 24.6 (21.2–28.3) | 15.3 (12.1–18.9) | 1.7 (0.1–7.5) |
| 20–29 | 22.7 (20.0–25.5) | 20.8 (20–21.7) | 52.7 (49.6–55.9) | 31.2 (26.3–36.3) | 11.6 (8.8–14.9) |
| 30–39 | 23.8 (21.3–26.3) | 27.5 (26.7–28.4) | 45.7 (42.6–48.8) | 27.8 (24.3–31.5) | 20.8 (17.7–24.2) |
| 40–49 | 19.3 (16.8–22.0) | 32.6 (31.7–33.6) | 34.4 (31.1–37.7) | 32.6 (28.9–36.4) | 27.3 (23.0–31.9) |
| **Marital or Union Status** | 0.86 | <0.001 | <0.001 | <0.001 | 0.132 |
| Never Married or Union | NA | 14.2 (13.6–14.9) | 38.0 (35.2–40.8) | 19.8 (16.8–23.1) | NA |
| Currently Married or Union | 22.0 (20.6–23.4) | 28.2 (27.6–28.9) | 41.3 (39.1–43.5) | 30.0 (27.1–33.0) | 20.3 (18.0–22.7) |
| Formally Married or Union | 23.4 (9.4–43.1) | 39.6 (35.1–44.3) | 71.0 (62.8–78.4) | 70.2 (47.5–87.6) | 10.2 (3.0–23.2) |
| **Place of Residence** | 0.281 | 0.005 | 0.70 | 0.253 | 0.018 |
| Rural | 21.5 (20.1–22.8) | 23.7 (23.1–24.2) | 40.9 (38.9–43.0) | 25.1 (22.4–28.1) | 22.3 (19.2–25.7) |
| Urban | 23.6 (19.7–28.3) | 22.0 (20.9–23.0) | 42.0 (36.7–47.5) | 27.9 (24.2–31.8) | 16.8 (13.8–20.2) |
| **Education** | <0.001 | <0.001 | <0.001 | <0.001 | 0.023 |
| Higher | 10.1 (6.5–14.6) | 14.6 (13.5–15.7) | 31.0 (27.2–34.9) | 18.7 (14.6–23.3) | 15.4 (11.8–19.5) |
| Secondary | 23.4 (20.4–26.7) | 20.2 (19.6–20.8) | 42 (39.4–44.5) | 24.8 (21.6–28.0) | 18.5 (15.0–22.4) |
| Primary | 19.4 (16.5–22.5) | 34.3 (33.0–35.6) | 46.7 (43.2–50.2) | 36.4 (32.2–40.8) | 24.6 (19.7–30.0) |
| No Education | 23.8 (21.7–26.0) | 37.9 (36.5–39.3) | 36.1 (28.3–44.5) | 37.8 (32.3–43.5) | 22.3 (18.1–26.9) |
| **Occupation** | 0.043 | <0.001 | <0.001 | <0.001 | <0.001 |
| Not working | 12.0 (7.7–17.4) | 11.1 (10.4–11.8) | 22.7 (19.0–26.6) | 17.6 (14.0–21.7) | 15.8 (6.2–30.6) |
| Professional/technical/managerial/services | 21.2 (19.0–23.9) | 22.7 (21.7–23.8) | 41.2 (38.5–43.9) | 24.6 (20.5–29.0) | 17.4 (14.2–20.9) |
| Agriculture | 21.8(19.5–24.2) | 27.2 (26.4–28) | 51.0 (47.0–55.0) | 27.8 (24.5–31.4) | 30.4 (25.5–35.5) |
| Skilled/unskilled manual | 23.6 (20.8–26.6) | 29.6 (28.6–30.6) | 48.0 (44.1–51.9) | 35.0 (30.9–39.2) | 17.7 (14.8–20.9) |
| **Wealth** | 0.384 | <0.001 | 0.024 | 0.005 | 0.030 |
| Poorest | 19.1 (17.0–21.4) | 29.1 (28–30.1) | 46 (42.0–49.9) | 34.0 (29.0–39.4) | 25.8 (20.5–31.7) |
| Poorer | 21.8 (19.4–24.3) | 26.8 (25.8–27.9) | 42.2 (38.7–45.8) | 27.9 (24.4–31.6) | 20.6 (15.7–26.2) |
| Middle | 21.6 (18.4–25.0) | 23.5 (22.6–24.3) | 37.9 (34.8–41.1) | 27.3 (23.5–31.2) | 18.4 (14.5–22.7) |
| Richer | 23.5 (19.6–27.7) | 21.0 (20–22.1) | 40.9 (36.4–45.5) | 27.2 (21.5–33.4) | 21.8 (16.8–27.4) |
| Richest | 23.8 (19.2–29.0) | 17.5 (16.2–18.8) | 37.3 (31.0–44.0) | 21.3 (17.4–25.6) | 14.8 (11.0–19.3) |
| **Access to the information** | | | | | |
| **Reading newspaper or magazine** | 0.004 | <0.001 | 0.076 | 0.603 | 0.489 |
| Not at all | 23.5 (21.8–25.2) | 30.5 (29.6–31.3) | 44.3 (40.8–47.8) | 28.2 (25.7–30.7) | 19.8 (16.9–23.0) |
| Less than once a week | 18.7 (14.5–23.3) | 22.5 (21.4–23.6) | 39.6 (35.6–43.7) | 26.5 (23.6–29.4) | 18.4 (13.9–23.6) |
| At least once a week | 15.4 (11.3–20.1) | 18.9 (18.3–19.7) | 39.9 (37.3–42.6) | 25.2 (17.2–34.4) | 21.9 (18.4–25.6) |
| **Frequency of watching television** | 0.016 | <0.001 | <0.001 | 0.115 | 0.676 |
| Not at all | 19.8 (17.8–21.9) | 26.3 (25.2–27.4) | 47.2 (41.5–53.0) | 29.2 (25.1–33.6) | 18.8 (15.0–23.1) |
| Less than once a week | 25.9 (21.5–30.8) | 26.2 (25.0–27.4) | 47 (42.6–51.3) | 29.0 (25.6–32.6) | 19.5 (14.6–25.1) |
| At least once a week | 23.3 (21.1–25.7) | 22.1 (21.5–22.7) | 39.5 (37.3–41.7) | 24.8 (21.0–28.9) | 21 (18.1–24.1) |
| **Frequency of listening radio** | 0.002 | 0.001 | <0.001 | 0.787 | 0.276 |
| Not at all | 18.9 (16.6–21.6) | 22.6 (22.1–23.2) | 43.8 (41.2–46.5) | 27.3 (24.0–30.9) | 20.3 (17.8–23.0) |
| Less than once a week | 24.7 (21.5–28.1) | 25.7 (24.2–27.2) | 39.1 (35.4–42.9) | 27.5 (24.8–30.2) | 16.5 (11.8–22.1) |

*(Continued)*

**Table 2.** (Continued)

| Factors | Afghanistan | India | Maldives | Nepal | Pakistan |
|---|---|---|---|---|---|
| | *P*–value | *P*–value | *P*–value | *P*–value | *P*–value |
| | Prevalence (%) | Prevalence (%) | Prevalence (%) | Prevalence (%) | Prevalence (%) |
| | (95% CI)[b] | (95% CI)[b] | (95% CI)[b] | (95% CI)[b] | (95% CI)[b] |
| At least once a week | 23.8 (21.6–26.1) | 23.2 (22.1–24.3) | 36.1 (32.9–39.4) | 26.1 (21.5–31.1) | 23.3 (16.9–30.7) |

[a] The data comes from Standard Demographic and Health Survey (DHS) carried out in Nepal in 2016, Pakistan in 2018, the Maldives in 2017, Afghanistan in 2015, and from National Family Health Survey conducted in India in 2016. The study population was men who were 15 to 49 years old. A complex survey design and sample weight were applied during analysis. The prevalence is expressed as a percentage, with a 95% confidence interval in parentheses. The chi–square test was used to get the p–value.

[b] Column percentages

NA.–Not Applicable

*Maldives*. The strength of association between cigarette smoking and age was the highest among the men aged 20–29 years old in the Maldives. The odds of being a cigarette smoker were 10% lower among men from poorer households than those from the poorest wealth status. The odds of cigarette smoking decreased with the increase in wealth index. Access to newspapers or magazines and radio also acted as protective factors for being a cigarette smoker in the Maldives. Men with lower than higher education levels were more likely to smoke cigarettes, while people who were employed in any work had a higher chance of taking cigarettes than those who were unemployed. The odds of cigarette smoking among men who lived in urban areas were 1.54 times than of those who lived in rural areas. Formally married men had a higher chance of smoking (AOR 3.19, 95% CI 2.16–4.71) than currently married men.

*Nepal*. Men with a lower educational level were more likely to smoke cigarettes in Nepal (AOR 2.70, 95% CI 1.83–3.99). The odds of cigarette smoking were 3.80 times higher among men who were formally married than men who were currently married. Men aged 20–29 years had the highest odds ratio among all age groups (AOR 2.62). In Nepal, living in an urban was also associated with cigarette smoking.

*Pakistan*. The chance of being a cigarette smoker increases with age in Pakistan. (AOR 23.25 among 40–49 years old men). A higher wealth index played a protective role of being a smoker; a significant association was found for the middle and richest wealth index with regard to cigarette smoking. Access to newspapers or magazines and television at least once a week was positively associated with cigarette smoking.

## Discussion

This study identified that the prevalence of cigarette smoking among men was higher in Maldives (41.2%) compared to the prevalence in Nepal (26.9%), India (23.0%), Afghanistan (22.0%), and Pakistan (20.1%). Our analysis showed that higher age, lower education, lower wealth index, and manual work were found as significant factors associated with smoking cigarettes among men in South Asian countries.

Age was found to be an important factor for cigarette smoking in the pooled data analysis. However, in the country-specific analysis, age was not significantly associated with cigarette smoking in Afghanistan. Men aged between 40 and 49 years had the strongest association with cigarette smoking in South Asian countries. A previous study in Ghana and India also found similar findings. Older men were more likely to smoke cigarettes compared to younger men in India [27] and Ghana [28]. Recently, lower smoking initiation among young people has been observed as the prevalence of smoking was lower than in past years [29–31]. Due to the cohort

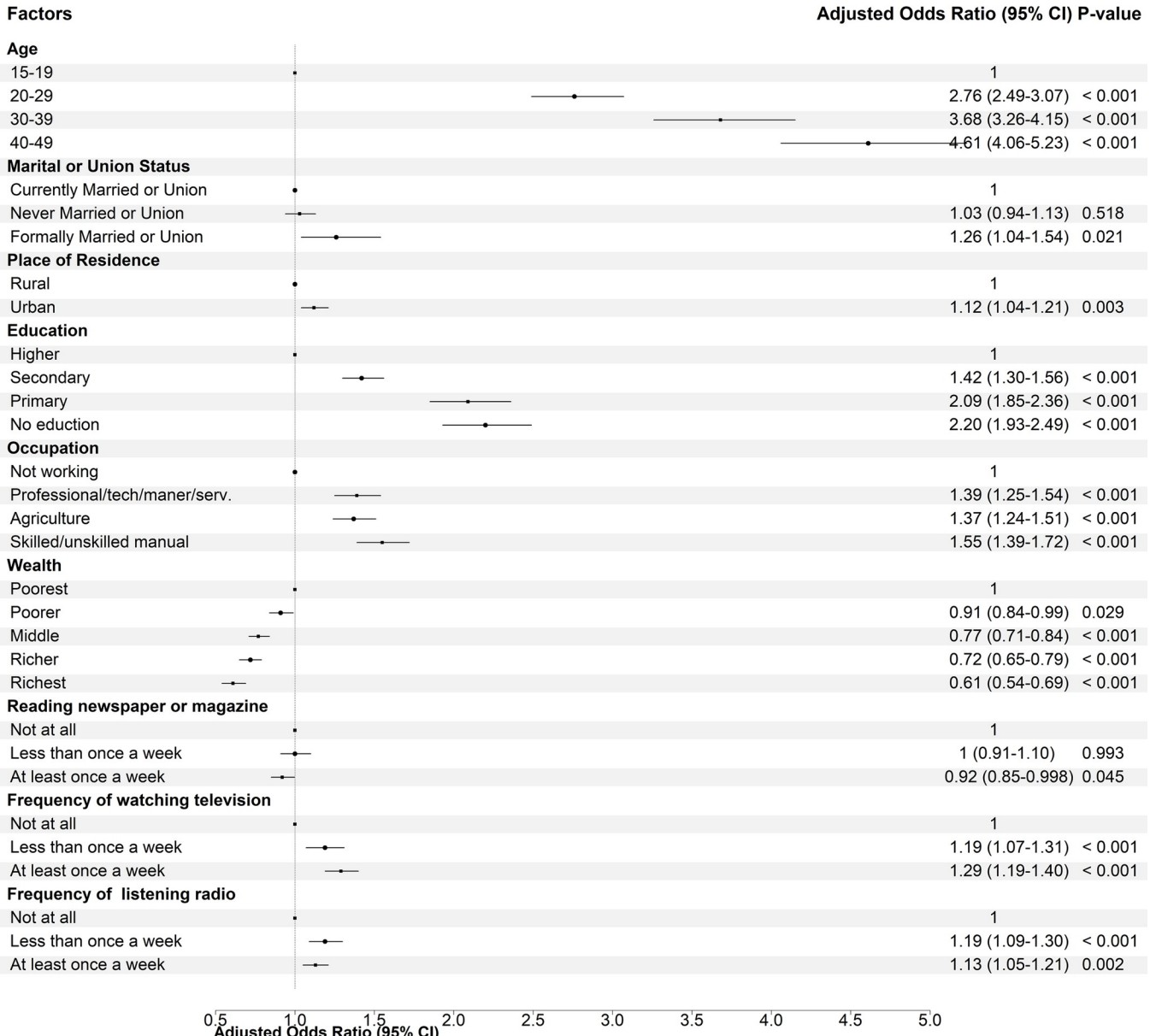

**Fig 4. Factors associated with cigarette smoking among South Asia men.** Multivariable logistic regression was performed on the pooled dataset from five south Asian countries.

effect and the prolonged trial time for cigarette consumption, the elderly in this region have a higher chance of smoking.

In the pooled analysis, we found that education played an important role in cigarette smoking in South Asia. In our country-specific model, the result indicated that men with no education or low level of education in all selected countries, except Pakistan, were more likely to be cigarette smokers. For example, in a study conducted in Srilanka and Malaysia, higher education was a protective factor against smoking [32, 33]. This can be explained that education raises awareness of the harmful effects of cigarette smoking on human health. Educated people may seek more smoking cessation support services and national quitline. In India, higher educated people tried to stop smoking more than lower educated people [18].

**Table 3. Multivariable logistic regression to identify factors of cigarettes in five South Asian countries[a].**

| Factors | Afghanistan | India | Maldives | Nepal | Pakistan |
|---|---|---|---|---|---|
| | AOR (95% CI) | AOR (95% CI) | AOR (95% CI) | AOR (95% CI) | AOR (95% CI) |
| **Age** | | | | | |
| 15–19 | 1 | 1 | 1 | 1 | 1 |
| 20–29 | 1.79 (0.93–3.44) | 2.86 (2.56–3.20)*** | 3.03 (2.33–3.94)*** | 2.62 (1.88–3.65)*** | 8.15 (1.17–56.81)* |
| 30–39 | 1.89 (0.99–3.60) | 3.58 (3.15–4.06)*** | 2.48 (1.82–3.38)*** | 1.90 (1.19–3.03)** | 16.34 (2.39–111.65)** |
| 40–49 | 1.42 (0.74–2.72) | 4.40 (3.84–5.03)*** | 1.49 (1.06–2.10)* | 2.44 (1.57–3.79)*** | 23.25 (3.37–160.55)** |
| **Marital or Union Status** | | | | | |
| Currently Married or Union | 1 | 1 | | 1 | 1 |
| Never Married or Union | NA | 0.972 (0.881–1.07) | 1.25 (1.02–1.54)* | 1.01 (0.754–1.37) | NA |
| Formally Married or Union | 1.29 (0.53–3.14) | 1.53 (1.23–1.90)*** | 3.19 (2.16–4.71)*** | 4.80 (2.08–11.1)*** | 0.417 (0.151–1.15) |
| **Place of Residence** | | | | | |
| Rural | 1 | 1 | 1 | 1 | 1 |
| Urban | 0.935 (0.567–1.54) | 1.22 (1.14–1.31)*** | 1.54 (1.10–2.15)* | 1.31 (1.05–1.64)* | 0.91 (0.639–1.29) |
| **Education** | | | | | |
| Higher | 1 | 1 | 1 | 1 | |
| Secondary | 2.47 (1.46–4.18)*** | 1.50 (1.36–1.64)*** | 1.81 (1.47–2.23)*** | 1.71 (1.35–2.15)*** | 1.10 (0.725–1.67) |
| Primary | 1.67 (0.90–3.09) | 2.15 (1.92–2.41)*** | 2.13 (1.62–2.80)*** | 2.49 (1.81–3.41)*** | 1.67 (0.996–2.80) |
| No education | 2.37 (1.35–4.14)** | 2.48 (2.18–2.82)*** | 1.65 (1.04–2.61)* | 2.70 (1.83–3.99)*** | 1.52 (0.89–2.60) |
| **Occupation** | | | | | |
| Not working | 1 | 1 | 1 | 1 | |
| Professional/technical/managerial/services | 1.67 (1.02–2.75)* | 1.40 (1.25–1.56)*** | 1.87 (1.41–2.48)*** | 0.98 (0.66–1.44) | 0.91 (0.32–2.57) |
| Agriculture | 1.63 (1.02–2.59)* | 1.38 (1.25–1.54)*** | 2.37 (1.74–3.24)*** | 0.96 (0.68–1.35) | 1.47 (0.53–4.10) |
| Skilled/unskilled manual | 1.94 (1.20–3.16)** | 1.69 (1.51–1.88)*** | 2.19 (1.63–2.94)*** | 1.32 (0.95–1.83) | 0.88 (0.329–2.33) |
| **Wealth** | | | | | |
| Poorest | 1 | 1 | 1 | 1 | 1 |
| Poorer | 0.961 (0.765–1.21) | 0.935 (0.868–1.01) | 0.908 (0.74–1.12) | 0.784 (0.59–1.04) | 0.724 (0.432–1.21) |
| Middle | 0.906 (0.699–1.17) | 0.817 (0.75–0.89)*** | 0.75 (0.592–0.951)* | 0.832 (0.597–1.16) | 0.563 (0.343–0.922)* |
| Richer | 0.945 (0.731–1.22) | 0.744 (0.676–0.818)*** | 0.682 (0.51–0.911)* | 0.849 (0.602–1.20) | 0.688 (0.402–1.18) |
| Richest | 1.03 (0.525–2.01) | 0.678 (0.596–0.771)*** | 0.526 (0.344–0.805)** | 0.583 (0.385–0.882)* | 0.43 (0.235–0.786)** |
| **Access to the information** | | | | | |
| **Frequency of watching television** | | | | | |
| Not at all | 1 | 1 | 1 | 1 | 1 |
| Less than once a week | 1.30 (0.945–1.77) | 1.11 (1.01–1.21)* | 1.04 (0.759–1.43) | 1.09 (0.803–1.49) | 1.31 (0.856–1.99) |
| At least once a week | 1.24 (1.01–1.51)* | 1.11 (1.03–1.20)** | 0.822 (0.623–1.08) | 1.07 (0.805–1.42) | 1.46 (1.02–2.10)* |
| **Reading newspaper or magazine** | | | | | |
| Not at all | 1 | 1 | 1 | 1 | 1 |
| Less than once a week | 0.747 (0.483–1.16) | 1.02 (0.931–1.12) | 0.774 (0.601–0.996)* | 1.06 (0.84–1.33) | 1.13 (0.765–1.68) |
| At least once a week | 0.567 (0.376–0.855)** | 0.905 (0.839–0.976)** | 0.769 (0.636–0.931)** | 1.22 (0.79–1.89) | 1.494 (1.003–2.23)* |
| **Frequency of listening to radio** | | | | | |
| Not at all | 1 | 1 | 1 | 1 | 1 |
| Less than once a week | 1.22 (0.97–1.53) | 1.26 (1.15–1.38)*** | 0.77 (0.65–0.92)** | 0.90 (0.71–1.12) | 0.8 (0.549–1.17) |

(*Continued*)

**Table 3.** (Continued)

| Factors | Afghanistan | India | Maldives | Nepal | Pakistan |
|---|---|---|---|---|---|
| | AOR (95% CI) | AOR (95% CI) | AOR (95% CI) | AOR (95% CI) | AOR (95% CI) |
| At least once a week | 1.19 (0.947–1.49) | 1.10 (1.02–1.19)* | 0.71 (0.588–0.857)*** | 0.799 (0.649–0.985)* | 1.25 (0.808–1.93) |

a Data were from Standard Demographic and Health Survey (DHS) conducted among men aged 15–49 years old in India in 2016, Nepal in 2017, Pakistan in 2018, the Maldives in 2017, and Afghanistan in 2015. A complex survey design and sampling weight were applied. Odds Ratio with 95% Confidence Interval in parentheses is shown.

*** p–value < 0.001

** p-value < 0.01

* p-value < 0.05

AOR- Adjusted Odds Ratio

NA–Not Applicable

Like previous studies [32, 34], we found that men who were working in any occupations were more likely to smoke cigarettes compared to those who were not working. Manual work was strongly associated with cigarette smoking in our pooled analysis. In country-specific analyses, manual work also increased the chance of cigarette smoking in Afghanistan, India, and the Maldives. Working individuals, particularly men, may experience work stress, which, in turn, may have a positive impact on being a cigarette smoker. An association between job strain and smoking intensity among men is well documented [35].

Living in urban areas was found to be statistically significant for cigarette smoking. More specifically, men in India, Maldives, and Nepal who came from urban areas were more likely to smoke cigarettes than men who come from a rural backgrounds. Urban people could more often be exposed to a smoking environment; thus, it changes people's smoking behaviour in this setting. Since the urban areas are more accessible to marketing, the tobacco industry can easily target the urban residents. A European study supports that people living in urban areas are more likely to smoke cigarettes [36].

Men with a high wealth index category had less chance to smoke cigarettes than other categories in India, Pakistan, and the Maldives. In Hungary [37] and Ghana [28], people with a low income also had a high probability of smoking cigarettes. Generally, people from lower economic status have less educational attainment, and they are more prone to have an addiction to drinking alcohol and tobacco smoking. They are also less informed about the dangers of smoking [29]. To find an in-depth explanation of the association between cigarette smoking and India's high wealth index, further studies should focus on the wealth index.

## Public health implication

The government and policymakers should design and implement interventions targeting high-risk groups to reduce the burden of cigarette smoking, which is a concern of the family and health professionals; briefly, the whole of society. Smoking cessation support services should be promoted in South Asian countries to accelerate quitting rate among elder groups. For example, a national toll-free quitline can be an effective service to assist people who want to quit smoking that is not available in Afghanistan, Nepal, Maldives, and Pakistan [37]. Effective public health education campaigns increase smoking quit ratios. The government should take an action plan to include strategies to be aware of the danger of cigarettes among students to prevent smoking. Moreover, universal access to education is found to be effective in reducing cigarette smoking. Because an educated person may have health education knowing that smoking is not a coping mechanism to deal with stress. Since there are a few studies conducted

focusing on factors of smoking cigarettes in South Asia, further research is needed to conduct, particularly a prospective cohort study with a long-term follow-up to establish a causal relationship between socio-demographic factors and cigarette smoking.

## Strengths and limitations

The key study strength is that it included a large sample size with high response rate data from nationally representative surveys, which generated sufficient power to examine the association between socio-demographic factors and cigarette smoking. Merging databases from five countries and pooled analysis enhanced statistical power and comparison of outcomes across different countries. Moreover, because the data were represented nationally with an adequate sample size and high response rate, this increases the precision and generalizability of the study. Therefore, the present findings could be generalized to similar socio-demographic characteristics and healthcare settings. However, the limitation of this study is that the data are from secondary sources, and the surveys were cross-sectional, so causal inferences could not be drawn. Another limitation is that we did not analyze data from all of the countries in South Asia. Data from Bangladesh, Bhutan, and Srilanka are missing. Also, data from Azad, Jammu and Kashmir, and Gilgit Baltistan of Pakistan were excluded due to unavailable cluster weight in the DHS dataset. The study population included men aged 15–49 years, which does not represent all age groups in South Asia. Besides, this self-reporting information collected was based on events, which may raise a possibility of recall bias. The prevalence of cigarette smoking might be underreported due to the conservative nature of people and social stigma.

## Conclusions

Our study findings imply men with higher age, low level of education, lower wealth status, urban residing, and manual workers were more prone to smoke cigarettes in the South Asian countries. Policymakers and public health practitioners should consider the identified factors for implementing effective interventions and country-specific programs to reduce smoking initiation and increase smoking cessation among men, while an improvement of smoking cessation support services can be an effective intervention in India, Nepal, Pakistan, Maldives, and Afghanistan.

## Supporting information

**S1 Table. Potential predictors of cigarette smoking among men in the South Asian countries in DHS and NHFS survey, 2015–2018.**
(DOCX)

**S2 Table. Country-wise prevalence of cigarette smoking among men in the South Asia.**
(DOCX)

**S3 Table. State-wise prevalence of cigarette smoking among men in the South Asia.**
(DOCX)

## Author Contributions

**Conceptualization:** Md Shariful Islam, K. M. Saif-Ur-Rahman, Arifuzzaman Khan.

**Formal analysis:** Md Shariful Islam.

**Investigation:** Md Shariful Islam.

**Methodology:** Md Shariful Islam.

**Project administration:** Mahbubur Rahman.

**Supervision:** Mamunur Rashid, Sarker Masud Parvez, K. M. Saif-Ur-Rahman, Arifuzzaman Khan.

**Visualization:** Md Shariful Islam.

**Writing – original draft:** Md Shariful Islam, Mamunur Rashid, Monaemul Islam Sizear, Raafat Hassan, Shuvon Chandra Hore.

**Writing – review & editing:** Md Shariful Islam, Raafat Hassan, Mahbubur Rahman, Sarker Masud Parvez, Rehnuma Haque, Farjana Jahan, Supta Chowdhury, Tarique Mohammad Nurul Huda, K. M. Saif-Ur-Rahman.

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
