## [Decision Letter · Decision Letter 0]

20 Jul 2022

PONE-D-22-09037Cigarette smoking and associated factors among men in five South Asian countries: a pooled analysis of nationally representative surveysPLOS ONE

Dear Dr. Md Shariful Islam

Thank you for submitting your manuscript to PLOS ONE. After careful consideration, we feel that it has merit but does not fully meet PLOS ONE’s publication criteria as it currently stands. Therefore, we invite you to submit a revised version of the manuscript that addresses the points raised during the review process.

We look forward to receiving your revised manuscript.

Kind regards,

Sonu Goel, MD

Academic Editor

PLOS ONE

Journal Requirements:

Reviewers' comments:

Reviewer's Responses to Questions

**Comments to the Author**

1. Is the manuscript technically sound, and do the data support the conclusions?

Reviewer #1: Yes

Reviewer #2: Yes

2. Has the statistical analysis been performed appropriately and rigorously? 

Reviewer #1: Yes

Reviewer #2: Yes

3. Have the authors made all data underlying the findings in their manuscript fully available?

Reviewer #1: Yes

Reviewer #2: Yes

4. Is the manuscript presented in an intelligible fashion and written in standard English?

Reviewer #1: Yes

Reviewer #2: Yes

5. Review Comments to the Author

Reviewer #1: The manuscript well written and easy to be followed

The section about public health implication needs to be expanded and provide specific country specific implication

The title of the figure should be under not above it.

Some figures have no titles

Reviewer #2: The study is apt and stands to contribute the the existing body of knowledge. It is however important that following revision are made

Introduction

1. Region in line 59 should be regions

2. The sentence on lines 59 - 61 should be revised particularly "The World Health Organization (WHO) set" is not clear

Methods

1. It appears that sample size for this study was not estimated which could have provided the basis for generalization and drawing of statistical inference. It will be good if the authors can provide the needed information in this regard in as much this study had used secondary data abstraction.

2. The rationale for using two different data sources (DHS and NHFS) is not clear and it will be good if a follow up information on this is provided in the manuscript. Additionally since these selected enumeration areas have different number of households, it would have been more scientific to have used proportion to size technique to determine number of households to be selected rather than picking fixed number of households per enumeration areas. However, provision of the rationale for the use of fixed number households by the authors will suffice.

3. The authors will need to provide more clarification of what the secondary data abstraction from DHS and NHFS were used for and that of the face to face interview. Additionally, the source of data collection tool used for face to face

interview should provided as well as how the reliability and validity of the tool were ascertained?

4. The authors have not included the criteria for inclusion in the study. It will be good if the inclusion and exclusion criteria used are clear stated. Additionally, the study focused on males only but females also engage in tobacco use, the rationale for excluding female may be needed.

5. The authors had stated that face to face interviews were conducted, however, it is unclear what that was used to achieve. Additionally, since there is a face to face component of the data collection, it is important that details of how consent for participation was obtained and also ethical approval may be required as tis is different from the DHS

Results

1.It is unclear how NA was taken care off or adjusted for in the analysis on table 2 and 3?

6. PLOS authors have the option to publish the peer review history of their article (what does this mean?). If published, this will include your full peer review and any attached files.

Reviewer #1: **Yes: **Huda Omer Basaleem

Reviewer #2: **Yes: **Tolulope Olumide Afolaranmi

---

## [Author Response · Author response to Decision Letter 0]

5 Aug 2022

Thank you for reviewing the manuscript critically. Your comments are valuable to improving the quality of the manuscript. We have updated the manuscript based on your comments. 

Reviewer 1

The manuscript well written and easy to be followed

The section about public health implication needs to be expanded and provide specific country specific implication

Response: Thank you for your compliments. 

The title of the figure should be under not above it.

Some figures have no titles

Response: Thank you for this comment. We replace the title of Table 1,2,3 from above to the bottom of the tables. Please see at page #8 line #171, page #13 line #203, and page #17 line #258. We also added the title of Figures 1, 2,3, and 4. Please see pages #32-35. We also replace the position of title in the supplementary file. Please see pages #1,2 and 3. 

Reviewer 2

The study is apt and stands to contribute the the existing body of knowledge. It is however important that following revision are made

Introduction

1. Region in line 59 should be regions

Response: Change has been made accordingly, see in line #56.

2. The sentence on lines 59 - 61 should be revised particularly "The World Health Organization (WHO) set" is not clear

Response: Thank you for your comments. We have updated the sentence. Please see lines 58-60. We removed the word "set ". It seems redundant. Now, the sentence is " The World Health Organization (WHO) introduced a target of a 25% reduction of death among individuals aged 30–70 years from cardiovascular diseases, diabetes, cancer, and chronic respiratory diseases between 2010 and 2025 ".

Methods

1. It appears that sample size for this study was not estimated which could have provided the basis for generalisation and drawing of statistical inference. It will be good if the authors can provide the needed information in this regard in as much this study had used secondary data abstraction.

Response: Thank you for your insightful comment. The datasets we merged were national representative for urban and rural areas, and for the first administrative level subdivisions, district data, and the response rate was quite high for each country (approximately 92% on average). The DHS research team calculated the sample size. Sample sizes for these DHS surveys are based on the number of survey domains (usually subnational units such as regions).. All survey sampling strategies are subject to sampling error. The DHS Program designs samples to provide national and subnational estimates with a reasonable relative standard error. We included all samples in our data analysis. As the data were represented nationally, even representative for the first administrative level, we think that there is no problem with the external validity (generalizability) of this study. Moreover, the high response rate and adequate sample size of each country provided sufficient statistical power, which abetted us to draw statistical inferences right. This has been clarify in the Method section page#4-5, lines#96-101, and in Discussion page#19, lines# 327-330. 

2. The rationale for using two different data sources (DHS and NHFS) is not clear and it will be good if a follow up information on this is provided in the manuscript. Additionally since these selected enumeration areas have different number of households, it would have been more scientific to have used proportion to size technique to determine number of households to be selected rather than picking fixed number of households per enumeration areas. However, provision of the rationale for the use of fixed number households by the authors will suffice.

Response: In India, DHS is known as NHFS. NHFS is the same as DHS. We have now updated line #86. About picking a fixed number of households per enumeration area, we agree with you that using the proportion to size technique to determine the number of households could be a better option. However, DHS picked a fixed number of households. The explanation from DHS is that a fixed number of households was selected to avoid the logistical burdens caused when the variable number of households is selected. At the same time, it is more cost-efficient. 

3. The authors will need to provide more clarification of what the secondary data abstraction from DHS and NHFS were used for and that of the face to face interview. Additionally, the source of data collection tool used for face to face interview should provided as well as how the reliability and validity of the tool were ascertained?

Response: We did not conduct any interviews. DHS surveys collect data through face-to-face interviews using households questionnaire incorporating reliable and valid tools and instruments. We’ve now revised our text related to a face-to-face interview in the manuscript, please see page #4, lines #99 – 100.

4. The authors have not included the criteria for inclusion in the study. It will be good if the inclusion and exclusion criteria used are clear stated. Additionally, the study focused on males only but females also engage in tobacco use, the rationale for excluding female may be needed.

Response: We revised and added inclusion and exclusion criteria on page 5, lines #104-107. We excluded women because the prevalence of cigarette smoking among women in South Asian countries is very low. In South Asia, women use mainly smoke-less tobacco, which is out of interest in this study.

5. The authors had stated that face to face interviews were conducted, however, it is unclear what that was used to achieve. Additionally, since there is a face to face component of the data collection, it is important that details of how consent for participation was obtained and also ethical approval may be required as tis is different from the DHS. 

Response: We did not conduct any interviews. We only analyse data from DHS. We think we were not clear enough hence the confusion. We revised our writing about face-to-face interviews. Please see page #5, lines #99, and 100. The DHS program took ethical approval to conduct surveys in Afghanistan, Nepal, India, Pakistan, and the Maldives. The DHS program also took written consent to collect data from participants. We took approval from the DHS program to perform this data analysis. Please see page#7, line #155-156 and page #20, lines #352-354. 

Results

1.It is unclear how NA was taken care off or adjusted for in the analysis on table 2 and 3?

Response: Among the five South-Asian countries we analysed data, the DHS in Afghanistan and Pakistan did not include men who are never married. Those countries included populations who were married or formally married. For this reason, the option became not applicable for these two countries. In our analysis, we used the currently married option as a reference value which is available for all countries. In our pool analysis, we re-normalised the s

---

## [Decision Letter · Decision Letter 1]

3 Nov 2022

Cigarette smoking and associated factors among men in five South Asian countries: a pooled analysis of nationally representative surveys

PONE-D-22-09037R1

Dear Dr. ISLAM,

We’re pleased to inform you that your manuscript has been judged scientifically suitable for publication and will be formally accepted for publication once it meets all outstanding technical requirements.

Kind regards,

Rajesh Raushan, PhD

Academic Editor

PLOS ONE

Additional Editor Comments (optional):

Not any

Reviewers' comments:

Reviewer's Responses to Questions

**Comments to the Author**

1. If the authors have adequately addressed your comments raised in a previous round of review and you feel that this manuscript is now acceptable for publication, you may indicate that here to bypass the “Comments to the Author” section, enter your conflict of interest statement in the “Confidential to Editor” section, and submit your "Accept" recommendation.

Reviewer #1: All comments have been addressed

Reviewer #2: All comments have been addressed

2. Is the manuscript technically sound, and do the data support the conclusions?

Reviewer #1: Yes

Reviewer #2: Yes

3. Has the statistical analysis been performed appropriately and rigorously? 

Reviewer #1: (No Response)

Reviewer #2: Yes

4. Have the authors made all data underlying the findings in their manuscript fully available?

Reviewer #1: Yes

Reviewer #2: Yes

5. Is the manuscript presented in an intelligible fashion and written in standard English?

Reviewer #1: Yes

Reviewer #2: Yes

6. Review Comments to the Author

Reviewer #1: No more comments. Authors had fully addressed my comments and the manuscript is suitable for publication

Reviewer #2: The authors have painstakingly effected all the review comments and hence the manuscript is publication worthy.

7. PLOS authors have the option to publish the peer review history of their article (what does this mean?). If published, this will include your full peer review and any attached files.

Reviewer #1: **Yes: **Huda Omer Basaleem

Reviewer #2: **Yes: **Tolulope Olumide Afolaranmi

---

## [Editor Report · Acceptance letter]

6 Nov 2022

PONE-D-22-09037R1 

Cigarette smoking and associated factors among men in five South Asian countries: a pooled analysis of nationally representative surveys 

Dear Dr. Shariful Islam:

I'm pleased to inform you that your manuscript has been deemed suitable for publication in PLOS ONE. Congratulations! Your manuscript is now with our production department. 

Kind regards, 

on behalf of

Dr. Rajesh Raushan 

Academic Editor

PLOS ONE